# Prognostic Role of Monocyte Distribution Width, CRP, Procalcitonin and Lactate as Sepsis Biomarkers in Critically Ill COVID-19 Patients

**DOI:** 10.3390/jcm12031197

**Published:** 2023-02-02

**Authors:** Dejana Bajić, Jovan Matijašević, Ljiljana Andrijević, Bojan Zarić, Mladena Lalić-Popović, Ilija Andrijević, Nemanja Todorović, Andrea Mihajlović, Borislav Tapavički, Jelena Ostojić

**Affiliations:** 1Department of Biochemistry, Faculty of Medicine, University of Novi Sad, Street Hajduk Veljkova 3, 21137 Novi Sad, Serbia; 2Institute for Pulmonary Diseases of Vojvodina, Put Dr Goldmana Street 4, 21204 Sremska Kamenica, Serbia; 3Faculty of Medicine, University of Novi Sad, Street Hajduk Veljkova 3, 21137 Novi Sad, Serbia; 4Department of Pharmacy, Faculty of Medicine, University of Novi Sad, Street Hajduk Veljkova 3, 21137 Novi Sad, Serbia; 5Department of Physiology, Faculty of Medicine, University of Novi Sad, Street Hajduk Veljkova 3, 21137 Novi Sad, Serbia

**Keywords:** monocyte distribution width, COVID-19, SARS-CoV-2, sepsis, biomarker, mortality

## Abstract

The severe acute respiratory syndrome coronavirus 2 (SARS-CoV-2) caused a global pandemic and one group of patients has developed a severe form of COVID-19 pneumonia with an urgent need for hospitalization and intensive care unit (ICU) admission. The aim of our study was to evaluate the prognostic role of MDW, CRP, procalcitonin (PCT), and lactate in critically ill COVID-19 patients. The primary outcome of interest is the 28 day mortality of ICU patients with confirmed SARS-CoV-2 infection and sepsis (according to Sepsis 3 criteria with acute change in SOFA score ≥ 2 points). Patients were divided into two groups according to survival on the 28th day after admission to the ICU. Every group was divided into two subgroups (women and men). Nonparametric tests (Mann–Whitney) for variables age, PCT, lactate, and MDW were lower than alpha *p* < 0.05, so there was a significant difference between survived and deceased patients. The Chi-square test confirmed statistically significant higher values of MDW and lactate in the non-survivor group. We found a significant association between MDW, lactate, procalcitonin, and fatal outcome, higher values were reported in the deceased group.

## 1. Introduction

The severe acute respiratory syndrome coronavirus 2 (SARS-CoV-2) caused a global pandemic health crisis by spreading rapidly across the world [1]. Coronavirus disease (COVID-19) is a highly contagious infectious disease that affects mostly the respiratory system [2]. Initially, the most common symptoms included fever, dyspnea, persistent cough, sore throat, nasal congestion, headache, muscle pain, and fatigue. Despite the fact that, in most cases, the disease has a mild form, one group of patients has developed a severe form of COVID-19 pneumonia with an urgent need for hospitalization and intensive care unit (ICU) admission [3]. Critically ill patients are defined as those with a high risk of imminent death accompanied by vital organ dysfunction [4].

During the COVID-19 pandemic infection, one of the biggest challenges for medical workers in the red COVID zones was critically ill patients prone to developing sepsis and septic shock. The life-threatening condition caused by the inappropriate host response to the infection is still the prevailing cause of morbidity, length of stay, and mortality within the ICU [5]. Intensive Care Units are highly equipped specialized wards established for close 24/7 monitoring of critically ill patients, rapid intervention, and extended treatment modalities. The Surviving Sepsis Campaign: International Guidelines for Management of Sepsis and Septic Shock 2021 brought updated recommendations for sepsis screening programs [6]. Sepsis screening tools are used to improve early recognition of sepsis and they include monitoring of vital parameters and estimation of diverse biomarkers associated with significant progression of infection to sepsis. According to the Sepsis 3 definition, the main criteria for setting a diagnosis of sepsis are suspected/confirmed infection and Sequential/Sepsis-related Organ Failure Assessment Score (SOFA score) of at least 2 points. SOFA core includes monitoring of six parameters PaO_2_/FiO_2_, number of platelets, estimation of bilirubin, creatinine, mean arterial pressure (MAP) and needs for dopamine, and Glasgow Coma Scale (GCS) score. Septic shock differs from sepsis due to more severe complications, and by definition, it involves persisting hypotension requiring vasopressors to maintain MAP ≥ 65 mmHg and lactate ≥ 2 mmol/L despite appropriate volume resuscitation. World Health Organization (WHO) has designated the finding of new diagnostic methods and biomarkers for early diagnosis of sepsis as the primary factor in treating sepsis, accompanied by the application of the relevant therapy [7]. Sepsis is a result of a multifaceted imbalance of proinflammatory and anti-inflammatory regulatory factors in an organism. An accurate, correct, and prompt diagnosis of imminent sepsis is a critical step in dealing with such complex conditions. Time is a crucial factor in treating sepsis in order to enable better overall survival. Sepsis itself, as a clinical syndrome, as well as its pathogenesis, have not been fully understood. Due to the presence of the non-specific signs and symptoms of the disease, its early detection and distinction from SIRS (Systematic Inflammatory Response Syndrome) are sometimes unclear [8]. That is why it is essential to set an accurate diagnosis of sepsis without delay by using the appropriate systems of measurement to assess vital parameters and biomarkers. According to the definition, prognostic biomarkers are used to identify the likelihood of a clinical event or disease progression in patients with certain medical conditions of interest [9]. Principally, prognostic biomarkers are key to identifying timely higher-risk populations for poor (fatal) outcomes. It is important to distinguish prognostic and predictive biomarkers (it indicates who will respond to specific therapy). The pinpointing of novel biomarkers applicable for the prediction of disease severity and outcomes is the main focus of research in patients with COVID-19 infection [10]. Preferably, these biomarkers should be easy to detect and useful for addressing clinical needs by clearly identifying whether a disease is presently active. Simple and affordable biomarkers should be obtained from routine laboratory analysis such as complete blood count and basic metabolic panel.

Besides the standard sepsis biomarkers like C-reactive protein (CRP), procalcitonin (PCT), and lactate, most frequently used in diagnostics and monitoring, we also assess a new cytometric parameter Monocyte Distribution Width (MDW). Monocytes represent a subset of white blood cells responsible for multiple activities as a part of the innate immune system [11]. Monocyte/macrophage population is directly involved in the immunopathogenesis of hyperinflammatory disorder in the severe form of COVID-19 [12]. Monocyte Distribution Width (MDW) represents a relatively novel cytometric parameter, automatically calculated along with complete blood count with differential (CBC-Diff) by the last generation of the DxH800 Hematology Analyzer (Beckman Coulter) [13]. It directly reflects the changes in the volume of monocytes affected by proinflammatory signals from the infective microorganisms connected with Pathogen Associated Molecular Patterns (PAMP) [14]. Monocytes, as the components of innate immunity, are involved in the response of the organism in the earliest phases of the infection, and they have multiple roles (phagocytosis, antigen-presenting cells, and the production of proinflammatory cytokine). Monocytes and their younger tissue-form macrophages work as professional phagocytes. Macrophages trigger coagulation and complement cascade. Sepsis is basically caused by immune disbalance due to hyperinflammatory syndrome [15]. The studies have determined that, a few hours prior to the clinical manifestations of sepsis, it is possible to detect the increase of MDW in blood corresponding to the intensity of the following reaction [16]. MDW provides useful information about changes in cellular volume and size of monocytes in peripheral blood associated with the current response to microbiological exposure [17]. Blood monocytes are the temporal phase between the place of production (bone marrow) and the place of action during infection (specific tissue) [18]. Hence, by assessing MDW and monocyte activation it is possible to consider the level and stage of infection. The changes in morphology and size correspond to the functional transformation of the monocytes, which represent the first line of defense against viruses [19]. Laboratory analysis plays a decisive role in the management of sepsis. Timely rapid diagnosis is essential because early treatment of life-threatening conditions can reduce mortality [20]. On the other hand, there are numerous biomarkers significant in practical clinical circumstances [21]. The challenge is to choose the right one for the appropriate purpose. The aim of our study was to evaluate the prognostic role of MDW, CRP, procalcitonin, and lactate in critically ill COVID-19 patients. The primary outcome of interest is the 28-day mortality of ICU patients with confirmed SARS-CoV-2 infection. The study was carried out in order to find out if there is a statistically significant difference between the survivors and deceased patients hospitalized in ICU regarding age and gender as well as to identify the additional influence of gender on the correlation between the different sepsis biomarkers in COVID-19 infection. Our research was focused on the prognostic importance of biomarkers in treating critically ill patients with COVID-19 infection and evaluating the outcome of the treatment.

## 2. Materials and Methods

We carried out a retrospective observational, single-center study between November 2020 and February 2022 on 160 patients admitted to the Intensive Care Units of the Institute for Pulmonary Diseases of Vojvodina in Sremska Kamenica in Serbia. The inclusion criteria were adult (>18 years) patients hospitalized in ICU, enrolled no more than once, with laboratory-confirmed COVID-19 infection (positive nasal rapid antigen test and reverse transcription polymerase chain reaction (RT-PCR) test conducted on a nasopharyngeal swab according to the WHO guidelines). Sepsis was diagnosed according to Sepsis-3 Criteria (confirmed infection with acute organ dysfunction using SOFA score ≥ 2 points). According to the outcome of the 28th day of hospitalization in the ICU, all patients were divided into two groups (survivors and deceased). The exclusion criteria for the trial were pregnant and breastfeeding women, patients who simultaneously have any malignant disease, AIDS, organ, or bone marrow transplantation, discharged from ICU in less than 4 h). The study was conducted in accordance with the amended Declaration of Helsinki and the International Conference on Harmonization (ICH) Good Clinical Practice (GCP) guidelines. The Ethics Committee of Institute for Pulmonary diseases of Vojvodina (Protocol Code No 9-II/3 on 24 February 2022) and the Ethics Committee of Faculty of Medicine University of Novi Sad approved this research (No 01-39/190/1, date of approval 13 May 2022).

We considered and analyzed all collected data from the patient’s medical records after admission to ICU (clinical symptoms and signs, all laboratory findings, age, gender, comorbidities, past medical history, course of disease during hospitalization, primary outcome—mortality rate at 28 days). Patients discharged alive or transferred prior to 28 days to another ward (with documentation about further treatment) were considered to be alive at 28 days.

According to available data from the medical records for each patient admitted to ICU standard laboratory, testing was performed to estimate the initial cross section of the overall health condition. Baseline assessment includes complete blood count with differential (CBC-Diff) and additional optional parameters for determination of MDW value. The whole human peripheral venous blood was collected routinely in sterile vacutainer tubes containing K2 EDTA (dipotassium ethylenediaminetetraacetic acid) for analysis of CBC and MDW (it is performed by automated hematology analyzer manufacturer Beckman Coulter, DxH 900, Nyon, Switzerland). CRP as a major positive acute phase reactant was diagnosed out of venous blood without anticoagulants by using the quality immunoturbidimetric method on the apparatus Cobas c311, made by Roche diagnostics, Basel, Switzerland. Procalcitonin was ascertained out of the serum using the apparatus Vidas 3 (manufacturer BioMerieux, Marcy-l Etoile, France). The principle of the test was to combine the immune-enzyme sandwich method with the final fluorescent detection (ELFA Enzyme-Linked Fluorescent Assay). Lactate was detected from plasma by using the apparatus Cobas c311 (manufacturer Roche diagnostics, Basel, Switzerland). The concentration of lactate was determined by the colorimetric measurement of the extinction at 660 nm using the commercial manufacturer’s set. All blood samples were taken within the first 24 h after admission to the ICU. These four parameters MDW, CRP, procalcitonin, and lactate were the main biomarkers of interest in the research.

Information about intrahospital 28-day mortality of critically ill patients with confirmed SARS-CoV-2 and sepsis admitted to ICU were regularly noted in medical documentation and so found.

### Statistical Analysis

Data were presented as mean and standard deviation for continuous variables (MDW, CRP, PCT, Lactate, age) and frequency (number and percentage%) for categorical variables (gender, outcome). SPSS (IBM SPSS Statistics for Windows, Version 26.0, Chicago, IL, USA) was used for all statistical analyses and tests. The significance level (α level) is set to 0.05 (5%), meaning that *p*-value < 0.05 is statistically significant and the null hypothesis is rejected. The normality test for all variables was conducted using the Shapiro–Wilk test. Those continuous variables that did not follow a normal distribution were expressed by means of medians and their interquartile range. To access differences between groups (survivors and deceased) for s specific variable, we used the non-parametric Mann–Whitney U test (comparing medians of a quantitative variable for the two categories of a dichotomous qualitative variable, e.g., men-women or survivors-deceased). The Chi-square test was used for multiple comparisons (two-way tables, e.g., women—survived and deceased, men—survived and deceased) for the categorical variables (with certain cut-off values of biomarkers). Cut-off values for MDW, CRP, PCT, and lactate were 26.0 μm, 100 mg/mL, 1 ng/mL, and 2 mmol/L, respectively. For each biomarker, the cut-off value separated the patients into two groups, those who had below and above cut-off values of certain biomarkers. Hence, in further statistical analysis, continuous numerical variables were shown as categorical variables, with 1 if the value of the biomarker was lower than the cut-off value, and 2 if the patient had the value of a biomarker higher than the cut-off. This method also enabled us to determine whether the association between two qualitative variables (men-women and survivors-deceased) is statistically significant. The Chi-square test is an overall test to detect relationships between two categorical variables. If the *p* value is lower than α level (0.05), the null hypothesis is rejected, so the difference between groups is statistically significant. The Kruskal–Wallis non-parametric test was used to compare three or more groups (continuous quantitative parameters) with one nominal variable. A *p* value < 0.05 had been considered significant. Relationships between variables were explored graphically (by means of box plots and scatterplots). Spearman rank correlation was done for nonparametric data to measure the degree of association between two variables.

## 3. Results

After a detailed assessment of all available laboratory data and medical history of COVID-19 patients in the ICU, 160 patients met the criteria for analysis. Exactly 54 patients (33.75% of total cases enrolled in the trial) belonged to the group that survived till the 28th day of hospitalization, among them were 23 women (42.6%) and 31 men (57.4%). The rest of the patients from the research, precisely 106 of them, were deceased (66.25%). Among non-survivors were 27 women (25.47%) and 79 men (74.53%) (Figure 1a,b). The sample size calculations were based on a 95% two-sided confidence interval.

Descriptive statistics of variables MDW, age, CRP, PCT, lactate, and ICU days are described as mean ± standard deviation (SD), minimum and maximum value, and interquartile range (IQR) (Table 1).

Test of normality (Shapiro–Wilk) for all parameters *p* < 0.05 showed a nonparametric distribution of data. The Chi-square test was used to compare differences between groups (survived and deceased on the 28th day), with cut-off values for MDW 26.0 μm, CRP 100 mg/mL, PCT 1 ng/mL, lactate 2 mmol/L respectively (Table 2).

This test showed that there is a statistically significant difference between survived and deceased patients for MDW. Higher value is noted in the deceased group of patients. For CRP there is no significant difference (*p* = 0.578), but lactate showed a statistically significant difference (*p* < 0.006), with higher values in the deceased group. On the other hand, the Chi-square test did not confirm a statistically significant distinction in PCT value (*p* = 0.08) between survivors and non-survivors for a cut-off value of 1 ng/mL. 

The Mann–Whitney U test was used to compare age, PCT, lactate, CRP, MDW, and ICU days, for certain outcomes (survived and deceased at 28 days) and results showed significant difference (alpha *p* < 0.05). The test showed a significant difference between survivors and non-survivors (*p* < 0.05) for certain variables age (*p* = 0.018), MDW (*p* = 0.008), PCT (*p* = 0.03), and lactate (*p* = 0.009) (Table 3). Additionally, length of stay in ICU (*p* = 0.704, 12.59 ± 12.12 days; min 1 day, max 82 days IQR 9 days) and CRP (*p* = 0.476) did not show statistically significant differences between groups (Table 3, Figure 2a,b).

Spearman’s correlation coefficient rho (Spearman rank correlation) is a non-parametric test used to measure the strength of association between two variables. The value of a coefficient between 0.4–0.69 means a moderate correlation, but rho between 0.7 and 0.89 represents a strong correlation. There is a moderate correlation between MDW and PCT for *p* < 0.05 (rho), ICU days and lactate, age and ICU, MDW and CRP, PCT, and CRP (Table 4.). Comparison between survived and deceased in average age, SOFA score, requirements for mechanical ventilation, and onset of shock in first 24 h is presented in Table 5. Receiver Operator Characteristic (ROC) curve analysis of MDW in predicting outcome showed AUC 0.627 (Figure 3).

## 4. Discussion

The World Health Organization emphasizes that the global annual mortality due to sepsis is around 6 million cases before the COVID-19 pandemic and a majority of them could be prevented with early recognition [22]. Therefore, WHO as an authority put tremendous effort to develop and implement new strategies for sepsis prevention, prompt diagnosis, and management. In present times, advances in the development of new biomarkers and combining existing ones to make effective indices and score systems are a priority [23]. The bridge between diagnostic methods and clinically experienced treatment of challenging diseases involves a good choice of laboratory analysis. SARS-CoV-2’s effortless transmission and transmutable nature caused an unprecedented global health crisis with rapidly increasing needs to cope with limited resources. Circulating monocytes are the first components of immunity to react proportionally to the intensity of the immune stimulus of the virus [24]. The monocytes in blood are the trajectory phase between the location of their creation in the bone marrow and the tissue they impact on [25]. The activation of the monocytes within the immune response is in fact estimated according to MDW, which is the direct measure of the intensity of the infection and an early indicator of the creation of sepsis [26]. This parameter illustrates the readiness of the immune defensive system. 

The well-timed detection of the life-threatening condition development is crucial. Each hour of delay at the beginning of the appropriate supportive medical therapy increases the mortality rate successively [6]. It is considered that the first 12 h in the early detection of sepsis and undertaking adequate therapeutic procedures are vital. That is why preliminary sepsis biomarkers like MDW indicate the infection a few hours prior to the strong reaction of the organism, warning medical workers to be alert and react in advance.

CRP is the most frequent marker of inflammation, but its biggest disadvantage is its low specificity due to the fact it is increased in all inflammatory processes, even when they are not caused by the infection [27]. Additionally, it is present in blood after the PAMP stimulus activity ceases. Procalcitonin is the commonest marker indicating sepsis, but its disadvantage is its high specificity, mainly in bacterial infections, while its use is limited by the price of its detection. FDA sepsis biomarker procalcitonin is accepted in practice and used in the assessment of antibiotic therapy de-escalation [28]. Its advantage is the ability to differentiate sepsis from non-infective origin SIRS, limiting relatively the unnecessary overuse and abuse of antibiotics. Furthermore, it contributes toward the decrease in multi-resistant bacteria development. CRP and procalcitonin are of protein structure and are created as the response to inflammation and/or infection [29].

Lactates are indicators of tissue hypoperfusion and are connected to multiorgan dysfunction and hypoxia. Their advantage is a good sensibility and low price of their detection, but their limitation is the increase in all phases of hypoperfusion and the fact that they are not typical to sepsis exclusively. The lactate value of over 2 mmol/L has been one of the criteria for the diagnosis of septic shock since 2016, along with the refractory hypotension of MAP < 65 mmHg, which indicates the use of vasopressor [30]. Within biochemical research, it is usual to monitor the levels of bilirubin and creatinine, bearing in mind they are non-specific parameters not capable of detecting sepsis solely [31]. For that reason, they are used for monitoring sepsis along with other results. There are numerous studies that confirm the connection between increased lactate and negative outcome of the treatment followed by the high mortality rate [32]. Recent studies proved that the hyperinflammatory response induced by SARS-CoV-2 is the main cause of the high mortality rate in infected patients.

Monocytes are known as the primary source of pro-inflammatory secretion biomolecules. In addition, MDW represents a window on the dynamics of monocyte activation status and preceding infection. Consequently, MDW could serve as a useful tool to guide clinical diagnosis and treatment of COVID-19 sepsis [14]. In one study, MDW < 20 has a high negative predictive value to exclude sepsis. This is especially important for clinicians to rule out suspicious diagnoses in unclear situations. In previous studies, MDW was reported to be a good sepsis biomarker alone or in combination with WBC, which enhances the diagnostic performance of diverse SIRS combinations [33].

The advantage of MDW as a parameter compared to the other ones is that it is detected along with the standard blood test results. Therefore, it can be detected by using a hematologic analyzer, because it does not require the additional blood testing of the patient applying the expensive reagents for its detection. Furthermore, it directly depicts the activities of the immune system caused by the infection in the organism exclusively. MDW can measure the change of volume and shape during the activation of monocytes, which makes this biomarker unique and exclusive in comparison to the other protein-based markers. It is specifically useful in the early detection of upcoming danger. Other target cells require several hours to start the production of certain proteins, which can be used as biomarkers. The MDW as a potential sepsis biomarker first was reported in 2017, and it is officially approved by FDA in 2019 [34].

The determination of cut-off values for continuous variables (sepsis biomarkers) in this research (MDW, CRP, PCT, lactate) was based on related literature and clinical experience in dealing with critically ill COVID-19 patients. Critical conditions—a severe form of COVID-19 and cytokine storm caused by high inflammation reaction of the host are followed by MDW values higher than 26.0 μm [35]. The severe disease course of COVID-19 was defined by CRP concentrations higher than 100 mg/mL, the requirement for oxygen therapy, and visible infiltrates on the chest X-ray [36]. Procalcitonin is a biomarker synthesized in all tissues during inflammation and it can be increased in a hyperinflammatory state in the absence of a bacterial pathogen. Due to the correlation between procalcitonin and hyperinflammatory state in critically ill patients, a severe form of COVID-19 infection could use 1 ng/mL as a cut-off value in prognosis [37]. In emergency and critical care medicine, lactate as an indicator of physiological stress and anaerobic metabolism is a useful parameter for the assessment of the severity of illness [38]. Some studies suggest using MDW in combination with white blood cells (WBC) is a better early sepsis indicator than MDW alone [26]. The diagnostic ability of MDW and WBC together was similar to CRP alone but better than PCT alone [26]. However, when MDW is higher than 21.5 μm, the addition of PCT increased the probability of sepsis. Patients with fatal outcomes had higher PCT, CRP, and lactate values than those that survived [27,28,29,37,38]. Results from our study imply that the estimation of MDW at admission in the first 24 h can easily detect patients with a higher risk of fatal outcomes. A limitation of this study is the relatively small sample size. In prospective new research, a larger series of patients should be considered to make appropriate conclusions.

## 5. Conclusions

Biomarkers can be useful for clinicians in the recognition and management of critically ill patients with COVID-19. Older age is associated with a greater risk of developing sepsis and a higher mortality rate. Delayed recognition of sepsis and treatment can lead to worse outcomes. MDW shows a significant correlation with inflammatory markers, and it could be used as an early indicator of sepsis. We found a significant association between MDW and final outcome, higher values are reported in the deceased group. It allows simple and fast monitoring of patients with COVID-19 infections and could be implemented in routine assessment.

## Figures and Tables

**Figure 1 jcm-12-01197-f001:**
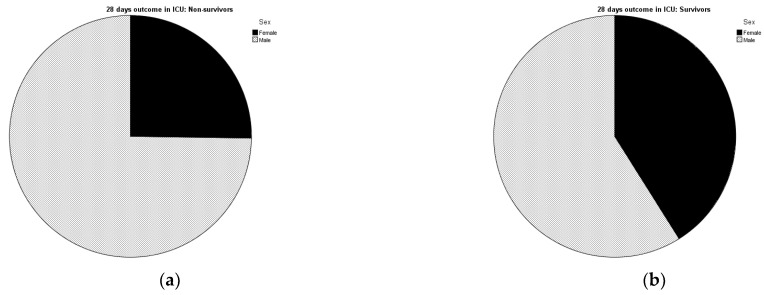
Pie charts showing percentage distribution of gender in group of (**a**) non-survivors and (**b**) survivors.

**Figure 2 jcm-12-01197-f002:**
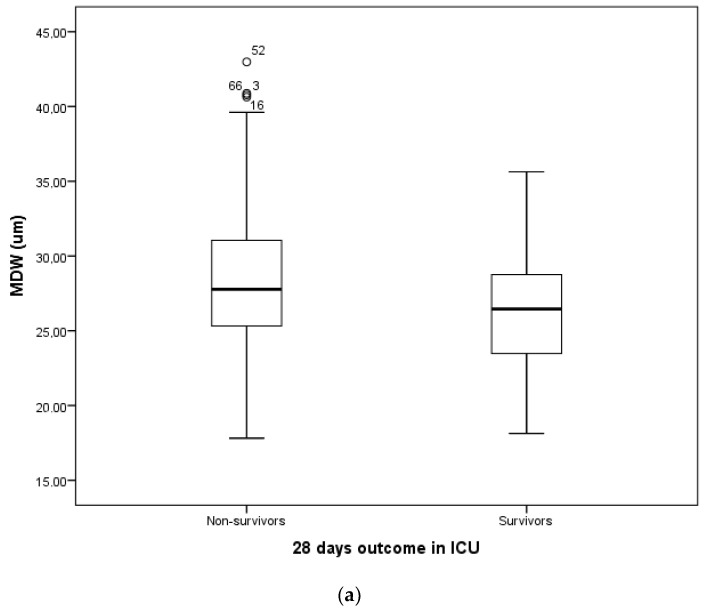
Box plots of (**a**) MDW (µm) and (**b**) CRP (mg/mL) in non-survivors and survivors group.

**Figure 3 jcm-12-01197-f003:**
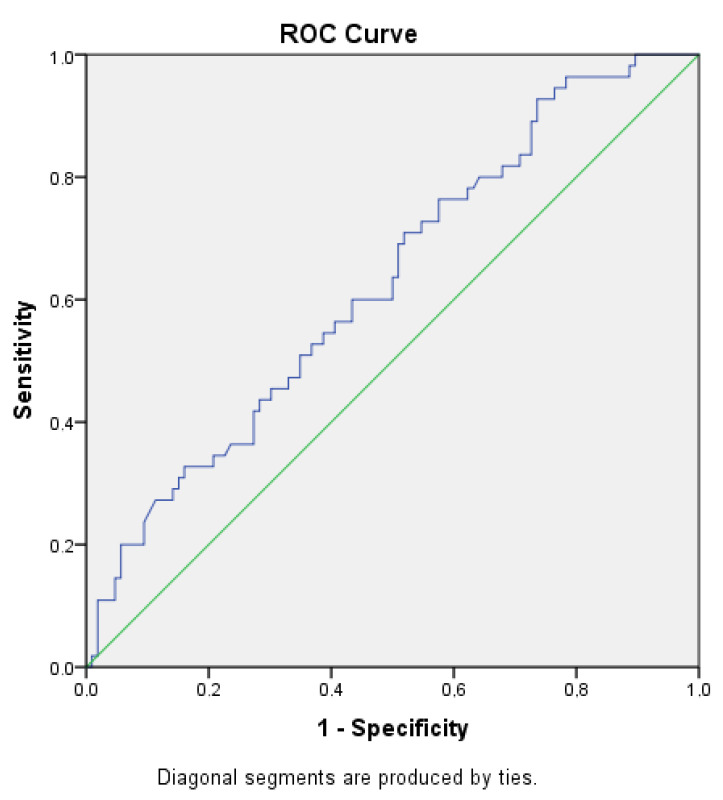
ROC analysis of MDW, AUC 0.627, blue line represent our data, green line represent uniformative test.

**Table 1 jcm-12-01197-t001:** Mann–Whitney U test was performed for group comparison (deceased and survived) and obtained *p* values are presented in the table. Descriptive statistics for selected variables (MDW, age, CRP, PCT, lactate, and ICU days) in certain groups are expressed as mean, standard deviation, minimum and maximum values, and interquartile range.

		*p* Value	Mean	SD	Min	Max	IQR
MDW	deceased	0.008	28.72	4.83	17.82	42.98	5.82
	survivors		26.46	3.78	18.13	35.63	5.47
Age	deceased	0.018	63.16	12.35	26	84	14
	survived		56.87	14.93	22	77	24.75
CRP	deceased	0.476	148.47	103.8	0.4	419.6	147.1
	survivors		132.46	80.6	0.5	320.7	136.05
PCT	deceased	0.03	2.12	6.7	0	61.44	1.04
	survivors		1.17	4.51	0.05	32.95	0.7
Lactate	deceased	0.009	3.13	2.9	1.02	19.75	1.36
	survivors		2.38	1.84	0.84	10.89	1.19
ICU days	deceased	0.704	11.92	11.23	1	82	12
	survivors		14.43	13.8	1	50	14.5

**Table 2 jcm-12-01197-t002:** Chi-square test for MDW, CRP, PCT, and lactate.

Biomarker	MDW	CRP	PCT	Lactate
Cut-off value	20 μm	100 mg/mL	1 ng/mL	2 mmol/L
*p* value, Chi-square test	0.023	0.578	0.08	0.006

**Table 3 jcm-12-01197-t003:** Man–Whitney U test.

Biomarker	Age	MDW	CRP	PCT	Lactate	ICU Days
*p* value	0.018	0.008	0.476	0.03	0.009	0.704

**Table 4 jcm-12-01197-t004:** Correlation between variables, * for *p* < 0.05; ** for *p* < 0.01.

		MDW	CRP	PCT	Lactate	ICU Days	Age
MDW	Correlation coefficient (r)	1.000	0.351 **	0.182 *	0.078	0.144	0.078
	Significance (*p*)	-	0.000	0.021	0.327	0.069	0.323
CRP	r	0.351 **	1.000	0.423 **	0.053	0.083	0.068
	*p*	0.000	-	0.000	0.499	0.292	0.391
PCT	r	0.182 *	0.423 **	1.000	0.087	0.060	0.017
	*p*	0.021	0.000	-	0.273	0.447	0.832
Lactate	r	0.078	0.053	0.087	1.000	0.167	0.207 **
	*p*	0.327	0.499	0.273	-	0.033	0.008
ICU days	r	0.144	0.083	0.060	0.167	1.000	0.306 **
	*p*	0.069	0.292	0.447	0.033	-	0.000
Age	r	0.078	0.068	0.017	0.207 **	0.306 **	1.000
	*p*	0.323	0.391	0.832	0.008	0.000	-

**Table 5 jcm-12-01197-t005:** Comparison between survived and deceased in average age, SOFA score, requirements for me-chanical ventilation, and onset of shock in first 24 h.

	Patients (Percentage)	Female	Male	Age (Mean)	SOFA Score (Mean)	Requirements for Mechanical Ventilation	Onset of Shock in First 24 h
Survived	54 (33.75%)	42.6%	57.4%	56.87	4.76	64.3%	14.3%
Deceased	106 (66.25%)	25.47%	74.56%	63.16	6.65	92.75%	29%

## Data Availability

Data are unavailable due to privacy and ethical restrictions.

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
