# Peer review of "Prognostic Role of Monocyte Distribution Width, CRP, Procalcitonin and Lactate as Sepsis Biomarkers in Critically Ill COVID-19 Patients"

_jcm, 2023, doi:10.3390/jcm12031197_

Round 1
Reviewer 1 Report
Manuscript
Prognostic role of monocyte distribution width, CRP, procalci-2 tonin and lactate as sepsis biomarkers in critically ill COVID-3 19 patients
by Dejana Bajić et al
Comments to the Authors
- the introduction is very long especially on the SARSCOV2 pandemic and little is dedicated to sepsis and its classification.
- Please clarify the and explain cutoff choices for all parameters and report the literature contribute that justify the choice
- the graphic in figure 1 and 2 need revision, I think the information should be report in 1 figure.
-".........distribution of data. Nonparametric tests (Mann- Whitneyev test) for variables age, PCT...........," 214 - must be correct
Author Response
Response to Reviewer 1 Comments
Thank you for taking a time for reviewing this manuscript and provide us with insightful suggestions.
Point 1: the introduction is very long especially on the SARSCOV2 pandemic and little is dedicated to sepsis and its classification.
Response 1: Our aim in detailed introduction about outbreak of COVID-19 pandemia was to highlight the most important facts about SARS-CoV-2 and diverse clinical manifestations of disease. The main field of our research was population of critically ill patients admitted to ICU, with confirmed COVID-19 disease and sepsis. We rewrote one part of manuscript with supplemented facts about sepsis and septic shock, Sepsis 3 definition and criteria for setting a diagnosis (confirmed infection and SOFA score≥2). We mentioned Surviving Sepsis Campaign: International Guidelines for Management of Sepsis and Septic Shock 2021 set of updated recommendations for sepsis screening tools. Page 2, rows 50-62
Point 2: Please clarify and explain cutoff choices for all parameters and report the literature contribute that justify the choice
Response 2: The choice of selected cut-off values for sepsis biomarkers (MDW, CRP, procalcitonin and lactate) reported in our clinical trial is based on clinical experience in ICU and relevant up to date literature and results from simililar studies about COVID-19 and sepsis. Page 11, rows 324-337.
-Cut-off for MDW 26μm literature (Riva G, Nasillo V, Luppi M, Tagliafico E, Trenti T. Linking COVID-19, monocyte activation and sepsis: MDW, a novel biomarker from cytometry. Lancet 2022;75:103754)
-Cut-off for CRP 100 mg/L literature (Esposito F, Matthes H, Schad F. Seven COVID-19 patients treated with C-Reactive protein (CRP) apheresis. Journal of Clinical Medicine 2022,11, 1956)
-Cut-off for procalcitonin 1ng/mL literature (Tong-Minh K, Does Y, Engelen S, Jong E, Ramakers C, Gommers D, et al. High procalcitonin levels associated with increased intensive care unit admission and mortality in patients with a COVID-19 infection in the emergency department. BMC Infectious Diseases 2022, 22, 165.
-Cut-off lactate 2 mmol/L literature (Bruno RR, Wernly B, Flaatten H, Fjolner J, Artigas A, Pinto BB. Lactate is associated with mortality in very old intensive care patients suffering from COVID-19: results from an international observational study of 2860 patients. Annals of intensive care 2021, 11, 128.
Point 3: the graphic in figure 1 and 2 need revision, I think the information should be report in 1 figure.
Response 3: We corrected mistake from picture 1 and 2, and save only picture 1 with appropriate explanation. Pie charts showing percentage distribution of gender among patients divided in two groups (survivors and non-survivors). Page 6, rows 206-211.
Point 4: ........distribution of data. Nonparametric tests (Mann- Whitneyev test) for variables age, PCT...........," 214 - must be correct
Response 4: Mann- Whitney U test was used to compare age, PCT, lactate, CRP, MDW for certain outcomes (survived and deceaseed at 28 day) and results showed significant difference (alpha p<0,05). Test showed significant difference between survivors and non-survivors (p<0,05) for certain variables age (p=0,018), MDW (p=0,008), PCT (p=0,03) and lactate (p=0,009) (table No. 3).
|
Biomarker |
Age |
MDW |
CRP |
PCT |
Lactate |
ICU days |
|
P value |
0.018 |
0.008 |
0.476 |
0.03 |
0.009 |
0.704 |
Reviewer 2 Report
First of all, thanks to the authors for this intensive study. However, we have some reservations about it indicated below.
1. The title of the paper says "...CRP, procalcitonin and lactate as sepsis biomarkers...", however the abstract and main text of the manuscript does not specify how the diagnosis of sepsis was made.
2. In the abstract method part, it should be written how many groups were in the study and how the groups were determined.
3. The abstract part the conclusion about MDW contained in the last sentence should be given in the abstract results section based on which findings were made (unlike other parameters).
4. In the main text method section, it should be stated on which device the CBC value is studied.
5. Parameters in the 1st column of Table 1 should be in the 1st row and p values should be written by making group comparisons.
6. The chi-square test for parametric data in Table 2 is not appropriate.
7. The Kruskal Wallis test performed in Table 3 is not suitable for comparison of two groups.
8. The p and r values should be stated in Table 4.
9. Instead of presenting pure literature in the discussion part, the data of the current study should be discussed in light of the literature.
10. As stated above, the paper contains important errors and should be rewritten.
Author Response
Response to Reviewer 2 Comments
Thank you for taking a time for reviewing this manuscript and provide us with insightful suggestions.
Point 1: The title of the paper says "...CRP, procalcitonin and lactate as sepsis biomarkers...", however the abstract and main text of the manuscript does not specify how the diagnosis of sepsis was made.
Response 1: We corrected part about how the diagnosis of sepsis is made. Sepsis was diagnosed according to Sepsis-3 Criteria (confirmed SARS-CoV-2 infection with acute organ dysfunction using SOFA score≥2 points). According to Sepsis 3 definition main criteria for setting diagnosis of sepsis are suspected/confirmed infection and Sequential/Sepsis -related Organ Failure Assessment Score (SOFA score) at least 2 points. SOFA core includes monitoring of 6 parameters PaO2/FiO2, number of platelets, estimation of bilirubin, creatinine, mean arterial pressure (MAP) and needs for dopamine, and Glasgow Coma Scale (GCS) score. These parameters are graded. Higher values of total SOFA score implies more serious complications and multiorgan dysfunction which means higher risk of fatal outcome.
We changed abstract and add what were criteria for sepsis
Point 2: In the abstract method part, it should be written how many groups were in the study and how the groups were determined.
Response 2: We corrected abstract: Patients were divided into two groups according to survival at 28th day after admisson in ICU (survived and deceased). Every group were divided into two subgroups (women and men). Biomarkers were analysed and compared among each group and subgroup.
Point 3: The abstract part the conclusion about MDW contained in the last sentence should be given in the abstract results section based on which findings were made (unlike other parameters).
Response 3: We changed conclusion part of abstract and involved all biomarkers considered in this research: We found a significant association between MDW, lactate, procalcitonin and final outcome, higher values are reported in deceased group
Point 4: In the main text method section, it should be stated on which device the CBC value is studied.
Response 4: We added information about device for CBC at page 4,rows 152-154 : The whole human peripheral venous blood was collected routinely in sterile vacutainer tubes containing K2 EDTA (dipotassium ethylenediaminetetraacetic acid) for analysis of CBC and MDW (it is performed by automated hematology analyser Beckman Coulter, DxH 900).
Point 5: Parameters in the 1st column of Table 1 should be in the 1st row and p values should be written by making group comparisons.
Response 5: We performed Mann-Whitney U test to make group comparison (deceased and survived) and present p vales for each variables (MDW, Age, CRP, PCT, Lactate, ICU days). In Table No I are presented Mean, standard deviation, minimum and maximum value and interquartile range for each variable.
|
|
P value |
Mean |
SD |
Min |
Max |
IQR |
|
|
MDW |
deceased |
0.008 |
28.72 |
4.83 |
17.82 |
42.98 |
5.82 |
|
|
survivors |
26.46 |
3.78 |
18.13 |
35.63 |
5.47 |
|
|
Age |
deceased |
0.018 |
63.16 |
12.35 |
26 |
84 |
14 |
|
|
survived |
56.87 |
14.93 |
22 |
77 |
24.75 |
|
|
CRP |
deceased |
0.476 |
148.47 |
103.8 |
0.4 |
419.6 |
147.1 |
|
|
survivors |
132.46 |
80.6 |
0.5 |
320.7 |
136.05 |
|
|
PCT |
deceased |
0.03 |
2.12 |
6.7 |
0 |
61.44 |
1.04 |
|
|
survivors |
1.17 |
4.51 |
0.05 |
32.95 |
0.7 |
|
|
Lactate |
deceased |
0.009 |
3.13 |
2.9 |
1.02 |
19.75 |
1.36 |
|
|
survivors |
2.38 |
1.84 |
0.84 |
10.89 |
1.19 |
|
|
ICU days |
deceased |
0.704 |
11.92 |
11.23 |
1 |
82 |
12 |
|
|
survivors |
14.43 |
13.8 |
1 |
50 |
14.5 |
|
Point 6: The chi-square test for parametric data in Table 2 is not appropriate.
Response 6: In table 2 are used non-parametric data, so chi-square test can be used to compare two categorical variables. Chi square test were used to compare differences between groups (survived and deceased at 28th day), with cut-off values for MDW 26.0 μm, CRP 100 mg/mL, PCT 1 ng/mL, lactate 2 mmol/L respectively.
Point 7: The Kruskal Wallis test performed in Table 3 is not suitable for comparison of two groups.
Response 7: We apologize for this mistake in terminology. Actually in Table No 3 is performed Mann-Whitney test and it is suitable for comparison of two groups. It was our mistake in the title of table No 3.
|
|
Point 8: The p and r values should be stated in Table 4.
Response 8: We marked correlation coefficient (r) and significance (p) for each variables MDW, CRP, PCT, Lactate, ICU days, Age in second column
|
|
MDW |
CRP |
PCT |
Lactate |
ICU days |
Age |
||
|
|
MDW |
Correlation coefficient (r) |
1.000 |
0.351** |
0.182* |
0.078 |
0.144 |
0.078 |
|
|
Significance(p) |
- |
0.000 |
0.021 |
0.327 |
0.069 |
0.323 |
|
|
CRP |
r |
0.351** |
1.000 |
0.423** |
0.053 |
0.083 |
0.068 |
|
|
|
p |
0.000 |
- |
0.000 |
0.499 |
0.292 |
0.391 |
|
|
PCT |
r |
0.182* |
0.423** |
1.000 |
0.087 |
0.060 |
0.017 |
|
|
|
p |
0.021 |
0.000 |
- |
0.273 |
0.447 |
0.832 |
|
|
Lactate |
r |
0.078 |
0.053 |
0.087 |
1.000 |
0.167 |
0.207** |
|
|
|
p |
0.327 |
0.499 |
0.273 |
- |
0.033 |
0.008 |
|
|
ICU days |
r |
0.144 |
0.083 |
0.060 |
0.167 |
1.000 |
0.306** |
|
|
|
p |
0.069 |
0.292 |
0.447 |
0.033 |
- |
0.000 |
|
|
Age |
r |
0.078 |
0.068 |
0.017 |
0.207** |
0.306** |
1.000 |
|
|
|
p |
0.323 |
0.391 |
0.832 |
0.008 |
0.000 |
- |
|
Point 9 Instead of presenting pure literature in the discussion part, the data of the current study should be discussed in light of the literature.
Response 9: We apologize for our detailed explanation in discussion part, we just wanted to connect all diverse perspectives on existing opinions and comprehension of COVID-19 infection and concrete biomarkers applicable in clinical practice. Prognostic role of these biomarkers, especially MDW which is relatively new parameter opens new possibilities for clinicians in coping with most serious health conditions at critically ill patients. Line between survivors and non-survivors is relatively thin; time and appropriate reaction to life-threatening state are crucial for better survival. Sepsis screening tools programmes are continuosly searching for simple, accurate, easy feasible biomarkers capable to distinguish patients in those with low and high risk of fatal outcome. Results from this research implies that there is significant difference between survivors and non survivors in sepsis biomarkers values. Despite the fact that numerous studies has analysed , biomarkers like CRP, procalcitonin and lactate, monocyte distribution width gives opportunity to recognize patients with COVID-19 and sepsis earlier than common biomarkers. Some studies suggest using of MDW in combination with white blood cells (WBC) as better early sepsis indicator than MDW alone [26]. Diagnostic ability of MDW and WBC together was similar to CRP alone, but better than PCT alone [26]. But, when MDW is higher than 21.5 μm, addition of PCT increased probability of sepsis. Patiens with fatal outcome had higher PCT, CRP and lactate values than survived [27,28,29,37, 38]. Besides all current attitudes toward frequent use of common sepsis biomarkers, results from our study implies that estimation of MDW at admission in first 24 hours can easily detect patients with higher risk of fatal outcome.
Reviewer 3 Report
The study is another attempt to evaluate a new biomarker in COVID-19 patients with sepsis. However, I think it must be better if certain corrections are made in the work.
1. I don't agree with the definition of a critically ill patient (39-40), why not ratio PaO2/FiO2...But, the Covid 19 patients is very complex and can damage other organs, not only the lungs
2. for each group of patient (survivors vs. death) should be prepared a table with demographic and clinical data (age, gender, SOFA score, therapy- tocilizumab, steroids, mechanical ventilation / invasive vs. noninvasive...)
3. Any serious study must take ROC curve analsys, used for assessing diagnostic tests and predictive models ( into the evaluation of MDW and clinical outcome)
4. Question, was the blood sample taken only once or not, was the trend of biomarker values monitored during the time (significant for PCT, CRP, lactate...)
5. In the discussion, compare your results with similar studies
5. I not seen ethical approval for the mentioned study (number, data, institution)
Thank you
Author Response
Response to Reviewer 3 Comments
Thank you for taking a time for reviewing this manuscript and provide us with insightful suggestions.
Point 1. I don't agree with the definition of a critically ill patient (39-40), why not ratio PaO2/FiO2...But, the Covid 19 patients is very complex and can damage other organs, not only the lungs.
Response 1: We corrected previous definition and stated general one in accordance with relevant literature (Towards definitions of critical illness and critical care using concept analysis. BMJ Open 2022;12e060972). Critically ill patients are defined as those with high risk of imminent death accompanied by vital organ dysfunction. Severity of disease is defined by respirstory rate > 30 breaths/min or SpO2 <90% or PaO2/FiO2 ratio ≤300 (WHO interim guidance for Covid)
Point 2. for each group of patient (survivors vs. death) should be prepared a table with demographic and clinical data (age, gender, SOFA score, therapy- tocilizumab, steroids, mechanical ventilation / invasive vs. noninvasive...)
Response 2: We compared SOFA score, requirement for mechanical ventilation,oneset of shock at first 24 hours after admission at survived and deceased. We estimated average age in each group, and distribution of gender among groups. Tocilizumab as monoclonal antibody against interleukin 6 receptor was applied only in strictly controlled indications, and in accordance with up to date Covid 19 guidelines. Interleukin 6 level was not routinely measured at patients. Corticosteroid therapy (methyprednisolone) was applied according to recommendations and WHO guidelines.
Table No 5 Comparison between survived and deceased in average age, SOFA score, requirements for mechanical ventilation and oneset of shock in first 24 hours
|
|
Patients (percentage) |
Female |
Male |
Age (Mean) |
SOFA score (Mean) |
Requirements for mechanical ventilation |
Oneset of shock in first 24 hours |
|
Survived |
54 (33,75%) |
42,6% |
57,4% |
56,87 |
4,76 |
64,3% |
14,3% |
|
Deceased |
106 (66,25%) |
25,47% |
74,56% |
63,16 |
6,65 |
92,75% |
29% |
Point 3. Any serious study must take ROC curve analsys, used for assessing diagnostic tests and predictive models ( into the evaluation of MDW and clinical outcome).
Response 3: As you suggested we performed ROC analysis for evaluation of MDW and outcome; Area Under the Curve is 0.627, confidence intervale 95% (0,538-0,717)
|
Area Under the Curve |
||||
|
Test Result Variable(s): MDW (um) |
||||
|
Area |
Std. Errora |
Asymptotic Sig.b |
Asymptotic 95% Confidence Interval |
|
|
Lower Bound |
Upper Bound |
|||
|
,627 |
,045 |
,008 |
,538 |
,717 |
|
The test result variable(s): MDW (um) has at least one tie between the positive actual state group and the negative actual state group. Statistics may be biased. |
||||
|
a. Under the nonparametric assumption |
||||
|
b. Null hypothesis: true area = 0.5 |
||||
Point 4. Question, was the blood sample taken only once or not, was the trend of biomarker values monitored during the time (significant for PCT, CRP, lactate...)
Response 4: Blood sample was taken more than once, but in our study we concerned values only at admission in ICU in first 24 hours. Due to epidemia of flu in Novi Sad, we are now not able to extract additional data from medical history of all patients, because entrance to Institute is strictly controlled.
Point 5. In the discussion, compare your results with similar studies
Response 5: We apologize for our detailed explanation in discussion part, we just wanted to connect all diverse perspectives on existing opinions and comprehension of COVID-19 infection and concrete biomarkers applicable in clinical practice. Prognostic role of these biomarkers, especially MDW which is relatively new parameter opens new possibilities for clinicians in coping with most serious health conditions at critically ill patients. Line between survivors and non-survivors is relatively thin; time and appropriate reaction to life-threatening state are crucial for better survival. Sepsis screening tools programmes are continuosly searching for simple, accurate, easy feasible biomarkers capable to distinguish patients in those with low and high risk of fatal outcome. Results from this research implies that there is significant difference between survivors and non survivors in sepsis biomarkers values. Despite the fact that numerous studies has analysed , biomarkers like CRP, procalcitonin and lactate, monocyte distribution width gives opportunity to recognize patients with COVID-19 and sepsis earlier than common biomarkers. Some studies suggest using of MDW in combination with white blood cells (WBC) as better early sepsis indicator than MDW alone [26]. Diagnostic ability of MDW and WBC together was similar to CRP alone, but better than PCT alone [26]. But, when MDW is higher than 21.5 μm, addition of PCT increased probability of sepsis. Patiens with fatal outcome had higher PCT, CRP and lactate values than survived [27,28,29,37, 38]. Besides all current attitudes toward frequent use of common sepsis biomarkers, results from our study implies that estimation of MDW at admission in first 24 hours can easily detect patients with higher risk of fatal outcome.
Point 6. I not seen ethical approval for the mentioned study (number, data, institution)
Response 6: The study was conducted in accordance with the Declaration of Helsinki, and approved by the Institutional Review Board and Ethics Committee of Institute for Pulmonary Disease of Vojvodina (Protocol Code No 9-II/3 on February 24th 2022) and Ethics Committee of Faculty of Medicine University of Novi Sad (No 01-39/190/1, date of approval May 13rd, 2022).

Round 2
Reviewer 2 Report
Thanks for your corrections, it is better now.
Reviewer 3 Report
Dear
After major revision of the study and adoption of all mentioned objections, I agree to publish the study.
Congratulations!